# LRRK2 and Lipid Pathways: Implications for Parkinson’s Disease

**DOI:** 10.3390/biom12111597

**Published:** 2022-10-30

**Authors:** Jasmin Galper, Woojin S. Kim, Nicolas Dzamko

**Affiliations:** 1Charles Perkins Centre and Faculty of Medicine and Health, School of Medical Sciences, University of Sydney, Camperdown, NSW 2050, Australia; 2Brain and Mind Centre and Faculty of Medicine and Health, School of Medical Sciences, University of Sydney, Camperdown, NSW 2050, Australia

**Keywords:** LRRK2, lipid, Parkinson’s disease, glucocerebrosidase, ceramide, BMP, glucose, metabolism, cholesterol, lysosome

## Abstract

Genetic alterations in the *LRRK2* gene, encoding leucine-rich repeat kinase 2, are a common risk factor for Parkinson’s disease. How LRRK2 alterations lead to cell pathology is an area of ongoing investigation, however, multiple lines of evidence suggest a role for LRRK2 in lipid pathways. It is increasingly recognized that in addition to being energy reservoirs and structural entities, some lipids, including neural lipids, participate in signaling cascades. Early investigations revealed that LRRK2 localized to membranous and vesicular structures, suggesting an interaction of LRRK2 and lipids or lipid-associated proteins. LRRK2 substrates from the Rab GTPase family play a critical role in vesicle trafficking, lipid metabolism and lipid storage, all processes which rely on lipid dynamics. In addition, LRRK2 is associated with the phosphorylation and activity of enzymes that catabolize plasma membrane and lysosomal lipids. Furthermore, LRRK2 knockout studies have revealed that blood, brain and urine exhibit lipid level changes, including alterations to sterols, sphingolipids and phospholipids, respectively. In human LRRK2 mutation carriers, changes to sterols, sphingolipids, phospholipids, fatty acyls and glycerolipids are reported in multiple tissues. This review summarizes the evidence regarding associations between LRRK2 and lipids, and the functional consequences of LRRK2-associated lipid changes are discussed.

## 1. Introduction

### 1.1. Overview

Missense mutations in leucine-rich repeat kinase 2 (LRRK2) are the most common risk factor for autosomal dominantly inherited Parkinson’s disease (PD) [1,2]. Polymorphisms in *LRRK2* are also associated with an increased risk of the more common sporadic form of PD [3]. Although the biological role of LRRK2 remains to be fully defined, studies have indicated that LRRK2 can be localized to membranous and vesicular structures, including lysosomes, endosomes, synaptic vesicles and mitochondria, suggesting that LRRK2 may have an affinity for lipids or lipid-associated proteins [4,5,6,7]. Indeed, increasing evidence from LRRK2 knockout rodents and LRRK2 mutation carriers further support a role for LRRK2 in processes relying on lipid membrane dynamics such as the endosome-lysosome system, the synaptic vesicle cycle, intracellular trafficking and lipid metabolism. This article reviews changes in lipids reported for subjects carrying LRRK2 mutations and LRRK2 animal models, and reviews the evidence regarding an association between LRRK2 and lipid-regulated processes known to be perturbed in PD.

### 1.2. An Introduction to Lipids

Lipids are commonly known for their roles as storage silos for energy, and for being important membrane components surrounding cells and organelles. However, it is increasingly recognized that some lipids, including neural lipids, are also bioactive signaling molecules that can directly facilitate signal transduction by means such as binding to receptors to initiate signaling cascades (reviewed in [8,9]). In contrast to proteins, lipids are not encoded by genes. Instead, lipids can be endogenously biosynthesized by enzymes using lipid precursors downstream of acetyl coenzyme A, a metabolite from glucose, fatty acid and amino acid catabolism [10]. Alternatively, some lipids are obtained from the diet [11]. The thousands of endogenous lipids that exist in humans have been organized into six categories based on their structure: glycerophospholipids, glycerolipids, sphingolipids, sterol lipids, prenol lipids and fatty acyls (Figure 1). Each category has a hierarchy of classes and sub-classes, which can be viewed online on the LIPID MAPS database (https://www.lipidmaps.org/data/classification/LM_classification_exp.php) [12]. With sensitive enough analytical methods such as mass spectrometry, information on the carbon chain length and double bond number (unsaturation) of lipid molecules are provided in following lipid nomenclature (extensively defined in [12]). For example, triacylglycerol 18:0/18:0/18:1 refers to a triacylglycerol with three carbon chains (separated by /), the first and second chain containing 18 carbons and 0 double bonds, and the third chain containing 18 carbons and 1 double bond (Figure 1). Lipid studies with and without such carbon chain and saturation resolution, which may have functional significance, are included in this review.

### 1.3. Lipid Alterations in LRRK2 Knockouts

Studies from rodents suggest that a loss of LRRK2 can result in both central and systemic changes in lipids (Figure 2) [13,14,15,16,17]. Sphingolipid changes found in the brains of sporadic PD patients [18] have spurred the interest in investigating potential lipid changes associated with PD risk genes such as *LRRK2*. In a small study examining brain sphingolipids via liquid chromatography-mass spectrometry (LC-MS), brains from LRRK2^-/-^ mice were found to have significantly increased levels of ceramide compared to WT (*n* = 3 in each genotype) [13]. The ceramide level comprised a summation of 13 long and very long chain ceramides, which were the only chain lengths detected. Levels of sphingomyelin, hexosylceramide (glucosylceramide/galactosylceramide) and sulfatide were unchanged, however, the authors noted that sphingomyelin and sulfatide from LRRK2^-/-^ brains trended to increase. Another targeted study specifically investigated a biomarker of lysosomal dysregulation, di-docosahexaenoyl (22:6) bis(monoacylglycerol) phosphate (di-22:6-BMP). Significantly decreased urine di-22:6-BMP was found in LRRK2^-/-^ mice compared to WT, as measured by LC-MS/MS [14]. LRRK2^-/-^ rats have also been found to have higher serum [15] and plasma [16] total cholesterol (cholesterol from low and high-density lipoprotein) compared to WT (Figure 2). In addition, LRRK2^-/-^ rats have lower total serum triacylglycerols compared to WT (ceramide and BMP were not investigated) [16]. The difference between total levels of lipid and specific lipid species is depicted in Figure 1. In summary, LRRK2^-/-^ rodents show changes in lipid levels from the sphingolipid, phospholipid, sterol and glycerolipid categories. The functional implications of lipid alterations are described in Section 2, Section 3 and Section 4 of this review.

### 1.4. Lipid Alterations in LRRK2 Mutation Carrier Humans

In contrast to LRRK2 knockouts, the main pathogenic PD-associated *LRRK2* mutations found in humans increase the kinase activity of LRRK2 [19,20], whereas loss-of-function *LRRK2* variants are not strongly associated with disease states in humans [21]. A metabolomics study found that LRRK2 R1441G PD patients had decreased plasma total cholesterol, whereas LRRK2 G2019S PD patients showed no change in cholesterol compared to controls, as assessed by MS (*n* = 8 participants per group) [22]. Whether this is linked to the R1441G mutation increasing the kinase activity of LRRK2 to a greater extent than the G2019S mutation remains to be elucidated [23]. Building on the findings that urine levels of di-22:6 BMP were increased in LRRK2^-/-^ mice, another two studies investigating phospholipids in urine via LC-MS/MS indicated that LRRK2 G2019S carriers showed significantly increased di-22:6 BMP and di-18:1 BMP compared to non-carriers [24] (Figure 2). Phosphatidylinositol (PI) 16:0/20:4 and PI 18:0/20:4 were also significantly increased in urine from the same cohort, while globotriaosylceramides were unchanged (*N* = 80). Significantly increased levels of di-18:1 BMP and di-22:6 BMP in urine were then validated in a second cohort (*N* = 116). One study also found these BMP isoforms were increased in LRRK2 R1441G/C carriers [25]. To better understand the extent to which lipid dysregulation occurs in LRRK2 mutation carriers, a recent, untargeted lipidomics study in LRRK2 G2019S carriers found significantly decreased levels of CSF hexosylceramide m38:0 (glucosylceramide/galactosylceramide), ceramide d32:1 and diacylglycerol 22:1e compared to non-carriers, as assessed by LC-MS/MS in *N* = 88 participants [26]. Whether such changes are reflected in the brains of LRRK2 mutation carriers remains to be elucidated. The same study reported serum lipid species from phosphatidylcholine, ceramide, sphingomyelin, phosphatidylethanolamine, monogalactosyldiacylglycerol, triacylglycerol and lysophosphatidylcholine sub-classes could significantly distinguish LRRK2 G2019S carriers from non-carriers in two separate cohorts of *N* = 221 and *N* = 315 participants [26] (Figure 2). Critically, this study demonstrated that in contrast to the total level of each lipid class changing in serum from LRRK2 G2019S carriers, different lipid species within each of these lipid classes were significantly altered in opposing directions of change [26]. One explanation is that lipid species within a class do not all behave the same and may even have inverse regulatory functions. For example, studies have found that very long chain ceramide exhibits an opposing effect on long chain ceramide in apoptosis [27,28,29]. However, the functional consequences of varying lipid length, saturation and linkage type are an area of development. Indeed, functional databases are mostly protein-centric, and the functional databases that include lipids (such as KEGG [30]) largely do not account for potential functional variation in lipids of varying unsaturation, chain length or linkage type. The functional consequences of phospholipid, glycerolipid, sterol, sphingolipid and fatty acyl LRRK2-associated changes are presented in Section 2, Section 3 and Section 4 of this review and lipid species-specific information is included where available.

**Figure 2 biomolecules-12-01597-f002:**
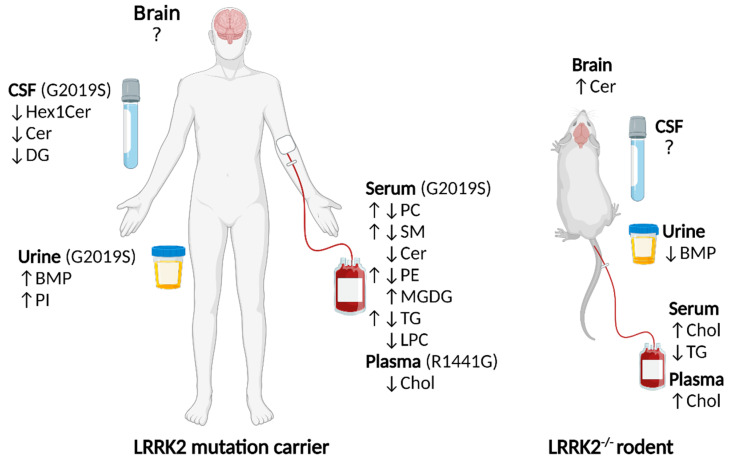
Lipid level changes in tissues from LRRK2 mutation carrier humans and LRRK2^-/-^ rodents. Human LRRK2 G2019S mutation carriers have alterations in serum phospholipid, sphingolipid and glycerolipid species, and CSF sphingolipid and glycerolipid species compared to those without a LRRK2 mutation [26]. Human LRRK2 G2019S mutation carriers have increased BMP and PI species in urine compared to those without the mutation [24]. Human LRRK2 R1441G mutation carriers have decreased plasma cholesterol [22]. LRRK2^-/-^ rats have higher serum [15] and plasma [16] cholesterol, and lower serum triacylglycerol [15] compared to WT. LRRK2^-/-^ mice have increased brain ceramide [13] and decreased urine BMP [14] compared to WT. For simplicity, specific lipid species are not depicted (detailed lipid species information is included in Section 1.3 and Section 1.4). Hex1Cer = hexosylceramide, Cer = ceramide, DG = diacylglycerol, BMP = bis (monoacylglycerol) phosphate, PI = phosphatidylinositol, PC = phosphatidylcholine, SM = sphingomyelin, PE = phosphatidylethanolamine, MGDG = monogalactosyldiacylglycerol, TG = triacylglycerol, LPC = lysophosphatidylcholine, Chol = cholesterol. Created with BioRender.com (accessed on 26 October 2022).

Triacylglycerol and cholesterol levels in the blood have been widely studied due to their relevance to cardiovascular disease [31]. In blood from pre-clinical or manifesting PD patients, the levels of “total” lipid such as triacylglycerol and cholesterol are controversial [32,33]. For example, one study on genotyped subjects reported that LRRK2-associated PD patients exhibited higher total serum triacylglycerol levels than other PD genotypes [34], while another study found no such difference [35]. Although the reason for this inconsistency is unclear, variability may arise due to unaccounted genotypic backgrounds that may influence lipid levels, for example, variations in apolipoprotein encoding genes such as *APOC2*, *APOA5* and *APOB* can influence triacylglycerol levels [36]. In addition, possible lipid level variability may arise from the capacity of analytical methods to detect different lipid species within the crude/total lipid level. Specifically, enzymatic and chemical assays can measure total lipid levels [37], whereas mass spectrometry techniques can provide more detailed information on specific lipid species of varying chain lengths and degrees of unsaturation, which may be important (Figure 1). In particular, information on lipid species can capture detail regarding levels of different species within a class changing in opposing directions. Future high-resolution studies in genetically well-characterized cohorts will aid in better understanding the nature of lipid alterations in PD. The majority of lipidomics studies have been conducted at the tissue or cellular level at one time point, however, sub-cellular studies would help clarify which organelles may exhibit lipid changes that reflect the tissue or whole cell changes found in LRRK2 studies. As evidence indicates that some intracellular lipids oscillate with the time of day [38], the sample collection time may be another covariate to consider in future study. 

To what extent peripheral changes in lipids are reflected in the brain depends on the lipid in question, as rates of peripheral exchange may differ across lipid types and in addition, lipids may be modified to cross the blood–brain barrier. For example, cholesterol is converted to hydroxycholesterol to pass across the blood–brain barrier [39]. Further, other brain lipids may be peroxidized to form metabolites such as aldehydes, which can be measured in the blood [40]. Future investigations measuring lipids and their associated derivatives, using matching brain and peripheral samples, could help clarify the concordance between brain and peripheral lipids.

## 2. LRRK2 and Lysosomal Lipid Metabolism

LRRK2 can be localized to lysosomes and LRRK2 mutations and inhibition of LRRK2 alter lysosomal function [41]. In addition, LRRK2 mutations result in enlarged lysosome size and a reduced number of lysosomes [42]. Studies on LRRK2^-/-^ rodent kidneys, revealed abnormal accumulation of lipofuscin [17,43,44], which is an aggregation of mainly protein and lipid, and to a lesser extent carbohydrate and metal [45]. Lipofuscin forms within lysosomes from undigested lysosomal material, and an accumulation of lipofuscin in LRRK2^-/-^ rodents may therefore indicate lysosomal dysfunction [17,43,44,45]. Genetic alterations to *LRRK2* also affect the lysosomal lipid BMP (also known as lysobisphosphatidic acid) [14,24]. BMP is localized in late endosomes/lysosomes and plays a role in the formation, structure and trafficking of endolysosomal compartments [46,47,48]. Interestingly, manipulation of BMP in cell lines and human fibroblasts indicates that BMP controls endolysosomal cholesterol levels [49,50] by enhancing the secretion of cholesterol-containing exosomes [50]. Another phospholipid which may be associated with genetic alterations to LRRK2 is phosphatidylinositol [24]. Phosphatidylinositol is a precursor to phosphoinositides, which are implicated in several lysosomal functions, including lysosomal cholesterol transport [51].

Intriguingly, recent studies [13,52,53,54,55] have pointed to a role for LRRK2 in regulating the lysosomal enzyme β-glucocerebrosidase (GCase), encoded by another common PD risk gene, *GBA1* [56,57,58,59]. GCase catabolizes the sphingolipids glucosylceramide and glucosylsphingosine into glucose and ceramide and glucose and sphingosine, respectively [60]. Mutations to GCase commonly associated with PD are loss-of-function, diminishing the enzyme’s hydrolytic activity [53,61]. GCase protein level is significantly decreased in LRRK2^-/-^ mouse brain [13] and human LRRK2 G2019S and I2020T frontal cortex [62]. Perhaps as compensation for reduced GCase protein levels, LRRK2^-/-^ mice brains, including striata, display significantly increased GCase activity compared to WT [13,52]. GCase activity from LRRK2 G2019S knock-in mice midbrain is also significantly higher compared to WT [55]. In human LRRK2 G2019S mutation carrier PBMCs [55] and fibroblasts [55], significantly increased GCase activity has been found compared to non-carriers, whereas a significant increase [53] and no change [63] in GCase activity have been reported in dried blood spots. A further study has indicated that GCase activity from dried blood spot samples decreases over time, which may explain reported inconsistencies [64]. In iPSC-derived dopaminergic neurons from LRRK2 mutation carriers (*n* = 2 G2019S and R1441C, *n* = 1 R1441G), significantly decreased GCase activity compared to controls (*n* = 2) has been reported, which could be restored upon treatment with a LRRK2 inhibitor [54]. However, another study using a different enzyme activity assay found significantly increased activity in dopaminergic neurons from LRRK2 G2019S mutation carriers (*n* = 2) compared to controls (*n* = 4) [55]. It has been suggested that the variability in the GCase results reported may be due to different GCase substrates utilized to measure enzyme activity, as commercial GCase substrates differ in their specificity for lysosomal and cytoplasmic GCase and sensitivity to other endogenous factors [55,65]. In addition, GCase activity may be affected in a tissue-specific manner [55]. Notably, inconsistencies may also arise from the low sample sizes available for iPSC-derived neuron investigations. Further large-scale studies separating lysosomal and total GCase readouts in situ will help clarify the nature of potential GCase activity alterations in LRRK2 mutation carriers. In addition, a mechanism behind a potential LRRK2-mediated effect on GCase warrants further exploration.

A key implication of LRRK2 affecting GCase activity is that LRRK2 could ultimately affect GCase substrates, specifically glucosylceramide and glucosylsphingosine breakdown. An alteration of GCase activity by LRRK2 could therefore result in altered metabolism or levels of ceramide, glucose, sphingosine, glucosylceramide and glucosylsphingosine in the lysosome. This altered lysosomal lipid metabolism may disrupt the lysosomal breakdown of proteins and other lipids, contributing to pathology.

In addition to being a metabolite of glucosylceramide breakdown, ceramide is also a metabolic product of sphingomyelin breakdown by acid sphingomyelinase in the lysosome [66]. Ceramide species are decreased in LRRK2 G0219S carrier CSF and serum and increased in LRRK2^-/-^ brains, while serum from LRRK2 G0219S carriers exhibits both increases and decreases in different sphingomyelin species (Figure 2). Whether potential alterations to serum, CSF and brain sphingolipids reflect alterations to lysosomal sphingolipid metabolism needs further investigation, as sphingolipids are found throughout the cell. Notably, however, mutations to *GBA1*, *GALC*, *SMPD1* and *ASAH1* have been linked to PD and all encode enzymes that catabolize sphingolipids in the lysosome [67,68,69,70]. Whether LRRK2 may directly or indirectly affect these enzymes and their ability to metabolize sphingolipids is an area that warrants further investigation. 

In summary, genetic alterations in *LRRK2* in humans and rodent models show consequences for lysosomal GCase [13,52,53,54,55] and endolysosomal BMP [14,24]. Lysosomal GCase catabolizes glucosphingolipids [61], while lysosomal BMP appears to control cholesterol levels [49,50]. In addition to BMP, the levels of cholesterol and sphingolipids are altered in LRRK2 mutation carrier humans and LRRK2^-/-^ rodents (Figure 2). Collectively, these data strongly support a role for LRRK2 in modulating lysosomal function, particularly the lysosomal metabolism of glucosphingolipids and sterols.

## 3. LRRK2, Mitochondrial Energy and Glucose/Insulin Pathways

### 3.1. LRRK2 and Insulin/Glucose

As mentioned previously, LRRK2 levels and activity affect the endolysosomal lipid BMP, impacting endolysosomal cholesterol levels. Experimentally induced elevation of cholesterol in cells and plasma via loss of the cholesterol transporters ABCA1 [71] or APOE [72], respectively, results in a marked reduction of insulin secretion and impaired glucose tolerance. A recent study also found that serum triacylglycerol containing 16:0 (palmitic acid) was a strong contributor to the differences between LRRK2 G2019S mutation carriers and non-carriers. Serum triacylglycerol comprising 16:0 from very low, low and intermediate density lipoprotein has previously been shown to positively correlate to an indicator of insulin resistance (HOMA-IR) and may be a better marker of insulin resistance than total serum triacylglycerol concentrations [73]. Notably, comparing serum lipid profiles of LRRK2 G2019S mutation carriers to non-carriers has revealed that lipids from insulin and glucose pathways are significant discriminators between the two groups, specifically, lipids involved in glycogen production, regulation of lipolysis in adipocytes, fat digestion and absorption (triacylglycerol and diacylglycerol species) and GLUT-4 translocation and glucose uptake (ceramide, diacylglycerol and triacylglycerol species) [26]. Recently, in CSF from LRRK2 G2019S mutation carriers, ceramide and diacylglycerol lipids, which are also implicated in glycogen/glucose/insulin pathways, were found to be significantly decreased by ~20% compared to non-carriers [26]. Further study is required to confirm these findings. In support of evidence regarding changes to glucose pathways in LRRK2-associated PD, the phosphorylation substrates of LRRK2 known as Rab GTPases have been implicated in the trafficking of the glucose transporter GLUT-4 to and from the plasma membrane. Genetic alteration to the LRRK2 substrates Rab5, Rab8a and Rab10 in muscle cells and adipocytes result in impaired insulin-stimulated translocation of GLUT-4 vesicles [74,75,76,77]. In addition, overexpression of the LRRK2 substrate Rab35 reverses a block of insulin-stimulated translocation of GLUT-4 in adipocytes [78]. Furthermore, fibroblasts from LRRK2 G2019S PD patients do not increase membrane GLUT-4 expression to the same extent as controls in response to insulin [79].

In vitro kinase assays demonstrate that LRRK2 can also phosphorylate synaptojanin-1 and in vivo, LRRK2 has been associated with increased synaptojanin-1 phosphorylation [80,81]. Synaptojanin-1 is a phosphoinositide phosphatase and importantly, mutations to the *SYNJ1* gene, encoding synaptojanin-1, are associated with Parkinsonism (symptoms of bradykinesia and either rest tremor, rigidity or both) [82,83,84]. Synaptojanin-1 deficiency results in impaired endolysosomal sorting and trafficking of multiple receptors and transporters including AMPA, transferrin and GLUT-1 [85]. Synaptojanin-1-deficient astrocytes display reduced expression of the glucose transporter GLUT-1, suggesting that synaptojanin-1 may disrupt glucose-sensing pathways and may contribute to glucose starvation [86]. Synaptojanin-1 dephosphorylates membrane phosphatidylinositol 4,5-bisphosphate [PI(4,5)P_2_] to form phosphatidylinositol (PI) [87]. It is speculated that the GLUT-1 membrane insertion by synaptojanin-1 involves PI(4,5)P_2_ hydrolysis and clathrin uncoating, as studies in cell lines indicate that a PI(4,5)P_2_-generating enzyme appears to inhibit the recycling of GLUT-1 from the endosome to the membrane [88]. Therefore, the involvement of LRRK2-associated synaptojanin-1 and LRRK2 Rab substrates in glucose transporter trafficking, together with LRRK2 mutation-associated changes in lipids involved in glycogen/glucose/insulin pathways, supports a role for LRRK2 in regulating glucose homeostasis.

### 3.2. LRRK2, Mitochondria and the Production and Storage of Energy

Interestingly, overexpression of WT LRRK2 in hepatic cell lines results in increased fatty acid oxidation, whereas LRRK2 knockdown decreases fatty acid oxidation. The effect on oxidation appears to be through LRRK2 regulation of the mitochondrial fatty acid transporter carnitine palmitoyltransferase 1A and the β-oxidation proteins peroxisome proliferator-activated receptor α and AMP-activated protein kinase. This suggests that LRRK2 plays a role in fatty acid catabolism [89]. In a recent untargeted lipidomics study of LRRK2 mutation carriers, serum lipids significantly discriminating LRRK2 G2019S mutation carriers from non-carriers belonged to pathways related to energy production, including mitochondrial oxidative phosphorylation (acylcarnitine, diacylglycerol and triacylglycerol species), glycerolipid metabolism (diacylglycerol, monogalactosyldiacylglycerol and triacylglycerol species) and retrograde endocannabinoid signaling to modulate mitochondrial respiration (phosphatidylcholine, phosphatidylethanolamine and diacylglycerol species) [26]. Additional lipid species found to be altered in LRRK2 G2019S mutation carrier serum are from lysophosphatidylcholine and phosphatidylcholine sub-classes [26]. These phospholipids are components of mitochondrial membranes. While phosphatidylcholine appears important for the structural integrity of both the outer and inner mitochondrial membranes [90], lysophosphatidylcholine increases mitochondrial membrane permeability to Ca^2+^, which ultimately affects oxidation rates [91]. As phosphatidylcholine is found in the plasma membrane and the membranes of other organelles, changes found in LRRK2 mutation carrier serum may reflect a perturbation to lipid membranes generally. Further, how LRRK2 may participate in such changes is an area requiring investigation.

Lipids can be stored in the form of lipid droplets, organelles comprising mainly of triacylglycerols and cholesterol esters, surrounded by a phospholipid monolayer. Lipid droplets are a reservoir of lipids for energy metabolism and membrane synthesis [92]. Multiple lines of evidence suggest that LRRK2 is involved in lipid droplet formation. For example, LRRK2^-/-^ rodents exhibit more lipid droplets in the kidney and liver than WT animals [15]. In humans, iPSC-derived dopaminergic neurons from LRRK2 G2019S mutation carriers have significantly more lipid droplets than controls [93]. Whether this is reflected in LRRK2 mutation carrier post-mortem brains remains to be elucidated. In addition, the LRRK2 substrate Rab8a has been shown to promote lipid droplet fusion and enlargement [20,94], which is further enhanced by increased Rab8a phosphorylation from the gain-of-function LRRK2 Y1699C mutation [20]. Another LRRK2 substrate, Rab10 [95,96], is necessary for recruiting autophagy machinery to engulf lipid droplets [97,98]. Rab10 is also involved in recruiting the accumulation of autophagy receptors on mitochondria to promote mitophagy [99]. Therefore, Rab10 is involved in the degradation of mitochondria and lipid droplets, both of which may have consequences for energy homeostasis. The LRRK2 substrate Rab5 [96] has been implicated in recruiting endosomes to lipid droplets, which possibly have a role in the transport of lipids from lipid droplets to the cell surface [100]. Therefore, LRRK2-Rab mediated changes in lipid storage may play a part in PD pathogenesis, potentially by a maladaptive use of energy supplies which could affect mitochondrial function or the ability of cells to replenish membrane lipids. In summary, alterations to LRRK2 appear to affect lipids and proteins involved in insulin and glucose signaling pathways, as well as pathways for storing and utilizing lipids for energy and membrane synthesis.

## 4. LRRK2 and Lipids in the Synaptic Vesicle Cycle

The regulation of lipid composition at synaptic membranes is essential for the proper functioning of the synaptic vesicle cycle, a process that relies heavily on lipid membrane dynamics. Although several candidate protein partners are reported for LRRK2 at pre- and post-synaptic sites (reviewed in [101]), those with roles in lipid metabolism or vesicle membrane turnover are presented below. Notably, a set of Rab GTPases are LRRK2 substrates that participate in synaptic vesicle trafficking. For example, the LRRK2 substrate Rab3 is involved in synaptic vesicle exocytosis, while and Rab5 and Rab35 are involved in vesicle recycling (for a review of Rab synaptic functions, see [102]). The expression of Rab5 at early endosomes present in presynaptic terminals supported the idea that Rab5 is involved in trafficking synaptic vesicles to either recycling or degradation [102,103,104,105]. It has been proposed that the endosome with Rab5 acts as a station for facilitating the turnover of aged lipid and protein components of synaptic vesicles [103]. The trafficking of vesicles by Rab5 is thought to occur through the recruitment of effector proteins which include phosphoinositide-3-kinase β, PI 4-phosphatase and PI 5-phosphatase. Through phosphoinositide-3-kinase β, Rab5 regulates the phosphorylation of PI(3,4,5)P_3_ from PI(4,5)P_2_ and subsequently its dephosphorylation to PI(3,4)P_2_ by 5-phosphatase and PI(3)P by 4-phosphatase [103,106,107,108]. The metabolism of PIs by Rab5 appears necessary for uncoating clathrin-coated vesicles, a necessary step in endocytosis [109].

Additional putative substrates subject to phosphorylation by LRRK2 are synaptojanin-1 [80] and endophilin A1 [110,111], which modulate synaptic vesicle endocytosis [87], a necessary step of regenerating synaptic vesicles after neurotransmitter release. Synaptojanin-1 is enriched at presynaptic terminals and its dephosphorylation of PI(4,5)P_2_ to PI at the presynaptic membrane appears to control vesicle recycling [87]. Specifically, it is thought that membrane PI(4,5)P_2_ is necessary for the recruitment of several endocytic proteins, including clathrin adaptors, and PI(4,5)P_2_ dephosphorylation to PI triggers the release of such clathrin adaptors, which is crucial for the presynaptic vesicle recycling [87,112,113,114]. LRRK2 G2019S [115] and synaptojanin-1 mutation mice [116] show a similar phenotype of accumulated clathrin coated vesicles and a reduced number of synaptic vesicles overall, indicating that disruption to either protein results in synaptic defects.

PI(4,5)P_2_ can also be hydrolyzed by phospholipase C to form inositol 1,4,5-trisphosphate (IP_3_) and diacylglycerol [117]. Priming and fusion of synaptic vesicles to the synaptic membrane required for exocytosis are regulated by diacylglycerol via its enhancement of Munc13-1 and protein kinase C activity [117,118,119,120]. In addition, membrane diacylglycerol can be phosphorylated by diacylglycerol kinases to form phosphatidic acid. Phosphorylation of membrane diacylglycerol is a necessary step for presynaptic vesicle retrieval [118]. Diacylglycerol 22:1e was reduced in LRRK2 G2019S mutation carrier CSF compared to non-carriers [26]. However, whether this indicates dysfunction of the synaptic vesicle cycle is unclear as notably, diacylglycerol is a second messenger in several diverse signaling pathways, including insulin/glucose signaling [KEGG map04931] and neurotrophic signaling [KEGG map04722]. Investigating a potential effect of LRRK2 on diacylglycerol at the level of the cell or organelle may help address the specific pathways affected by potential diacylglycerol changes.

Serum lysophosphatidylcholine and phosphatidylcholine species from LRRK2 G2019S mutation carriers differ from those without a LRRK2 mutation (Figure 2). These two lipids are metabolically linked through the enzyme calcium-independent phospholipase A2 (CaI-PLA2), which metabolizes phosphatidylcholine to lysophosphatidylcholine. Mutations to CaI-PLA2’s encoding gene, *PLA2G6*, are linked to Parkinsonism and PD [121,122,123,124]. Loss of the *PLA2G6* homolog in *Drosophila* reveals smaller synaptic vesicles than WT, suggesting an important role for these phospholipids in the synaptic vesicle cycle [125]. Interestingly, lysophosphatidylcholine, diacylglycerol and ceramide, which all have evidence regarding LRRK2-associated alterations (Figure 2), are described as cone-shaped lipids that promote membrane curvature, which is conducive of exocytosis [8,126]. To clarify whether these LRRK2-associated lipids may be indicative of synaptic vesicle changes, however, further studies in the CNS are required as currently, brain lipidomics studies in human LRRK2 mutation carriers are lacking.

As presented in Figure 2, genetic alterations in *LRRK2* are associated with altered levels of blood cholesterol in rodents [15,16] and humans [22]. How exactly LRRK2 may influence cholesterol is an area of investigation, however, it may involve the trafficking and metabolism of cholesterol in the lysosome, which is potentially mediated by BMP [49,50] and GCase [127], respectively. Cholesterol is the most abundant lipid in synaptic vesicles and influences membrane properties such as membrane curvature, fluidity and thickness, which affect fusion and exocytosis [128,129,130,131,132]. The concentration of cholesterol in membranes affects the clustering of SNARE (Soluble NSF Attachment Protein Receptor) proteins that define docking and fusion sites for exocytosis [129,133,134]. Cholesterol may also affect the anchoring of the LRRK2 substrate Rab3a [95,96,135], which is involved in a late step of synaptic vesicle exocytosis (reviewed in [136]). Another synaptic protein influenced by cholesterol is the pathological hallmark of PD, α-synuclein. The function of α-synuclein has been challenging to assess, in part because the secondary structure of the protein changes depending on whether it is membrane-bound or in the cytosol [137,138,139,140,141,142,143,144,145,146,147,148]. Evidence suggests that one role of α-synuclein involves the clustering of synaptic vesicles or vesicle-vesicle interactions [149,150,151,152,153]. In vitro evidence indicates that cholesterol promotes vesicle-vesicle interactions by altering the conformation of membrane-bound α-synuclein [154]. Studies in neuronal cell lines and primary human neurons overexpressing α-synuclein have demonstrated that statins, which inhibit sterol synthesis, reduce levels of aggregated α-synuclein [155]. This suggests that elevated cholesterol may contribute to synuclein aggregation, however, low cholesterol levels may disrupt the normal interaction of cholesterol and α-synuclein which are required for vesicle trafficking. Therefore, cholesterol dyshomeostasis arising from both excess or limited supply of cholesterol has the potential to result in neurological defects. Further studies are needed to confirm the report of decreased plasma cholesterol found in LRRK2 R1441G carriers, whether this applies to all LRRK2 gain-of-function mutations, and whether this may reflect an increase or decrease in intracellular cholesterol levels. Additionally, polyunsaturated fatty acids (PUFAs) have been implicated in α-synuclein oligomerization [156,157], however, whether PUFAs are affected by LRRK2-associated changes has not been clarified.

In summary, the LRRK2 substrates or putative substrates Rab5 and synaptojanin-1 affect the metabolism of PI, either indirectly or directly, respectively, to mediate endocytosis. Lysophosphatidylcholine, diacylglycerol, ceramide and cholesterol are altered in LRRK2 mutation carriers and may be involved in exocytosis, however, the diverse functions of these lipids mean that further studies are required to elucidate possible mechanisms of involvement in LRRK2-associated PD. The effects of LRRK2 involving lipid pathways are summarized in Figure 3. 

## 5. Lipid Biomarkers and Pharmacological LRRK2 Inhibition

Given that gain-of-function mutations in LRRK2 are associated with PD and altered lipid homeostasis, pharmacological inhibition of LRRK2 kinase activity holds therapeutic potential. For example, the endolysosomal phospholipid di-22:6-BMP, thought to be a biomarker of lysosomal dysregulation, is increased in urine from LRRK2 G2019S mutation carriers and is reduced in nonhuman primates treated with brain-penetrating LRRK2 kinase inhibitors (GNE-7915 and GNE-0877/DNL201) [14]. In phase 1 and phase 1b clinical trials, healthy human volunteers and PD patients treated with the LRRK2 inhibitor DNL201 had reduced urinary di-22:6-BMP concentrations [158]. In LRRK2 mutation carriers, the BMP alterations described previously in urine [24] were not additionally found in CSF [26], however, the BMP species detected in CSF did not include the those reportedly altered in urine. Therefore, whether the lack of reports regarding changes in CSF BMP is due to technical differences between studies or central versus peripheral differences is unknown. Another LRRK2 kinase inhibitor, BIIB122 (DNL151), which has apparently improved pharmacokinetic properties [158], is currently being tested in a phase 2 clinical trial (NCT05348785). Results from this study will indicate whether LRRK2 inhibitors improve clinical measures such as the Unified Parkinson’s Disease Rating Scale (UPDRS). Whether di-22:6-BMP correlates to such clinical measures is also of interest, as has been found by one study which indicated urinary di-18:1-BMP and total di-22:6-BMP were associated with a worse cognitive performance by the Montreal Cognitive Assessment (MoCA) [24]. Such lipid correlates of clinical progression may aid in the development of PD biomarkers.

## 6. Lipid Modulators in LRRK2 Models and Parkinson’s Disease

In a study that utilized a Food and Drug Administration (FDA)-approved drug library to screen for compounds that could rescue LRRK2 G2019S neurite degeneration in mice and flies, the statin lovastatin was found to prevent dopaminergic neuron loss. In addition, lovastatin rescued motor deficits in LRRK2 G2019S flies [159]. Statins inhibit cholesterol biosynthesis, however, they also have other effects like lowering coenzyme Q10 [160] and triacylglycerols and reducing inflammatory responses [161]. In addition, lovastatin has been shown to reduce α-synuclein pathology in PD models [155,162]. Although the evidence reviewed in this article suggests that cholesterol and triacylglycerols may be perturbed in LRRK2-associated PD, how potential neuroprotection in PD is conferred is an area of investigation. Recently, results from a lovastatin phase 2 clinical trial in non-genotyped PD patients demonstrated that those on lovastatin had significantly less deterioration in the striatal dopamine uptake ratio on positron emission tomography (PET) imaging compared to placebo, although UPDRS scores were not significantly altered in the 48-week period [163]. However, given the marked effect of lovastatin on striatal dopamine uptake observed, larger follow-up studies are warranted. Additionally, investigating a potential variable impact of statins on the LRRK2 genotype is of interest.

## 7. Conclusions

In summary, a collection of evidence indicates that LRRK2 phosphorylates substrates that either directly metabolize phospholipids in the synaptic membrane (synaptojanin-1) or recruit enzymes that do (Rab5). These effects appear essential for the proper functioning of the synaptic vesicle cycle. In addition, LRRK2 appears to affect GCase, which metabolizes sphingolipids in the lysosome. Furthermore, the LRRK2 substrates Rab5, Rab8 and Rab10 are involved in lipid trafficking, lipid droplet fusion and lipophagy, respectively. Rabs are additionally involved in trafficking receptors such as GLUT-4, critical in insulin and glucose pathways. Future investigations should address the variability encountered in LRRK2-GCase interaction studies and clarify the potential mechanisms by which this occurs. Although the LRRK2 knockout models have been informative regarding potential LRRK2-associated lipid alterations, future investigations on LRRK2 mutation knock-in models may provide a more comparative model of human LRRK2 mutation changes. In addition, further studies in brains from humans that carry LRRK2 mutations will help clarify whether peripheral and CSF lipid changes are reflected in the brain. LRRK2 kinase inhibitors and statins have lipid-altering effects, however, whether and how this benefits PD patients are an area of active investigation. In addition, whether the unique lipid signature of LRRK2 mutation carriers has diagnostic and prognostic biomarker potential is of interest. 

## Figures and Tables

**Figure 1 biomolecules-12-01597-f001:**
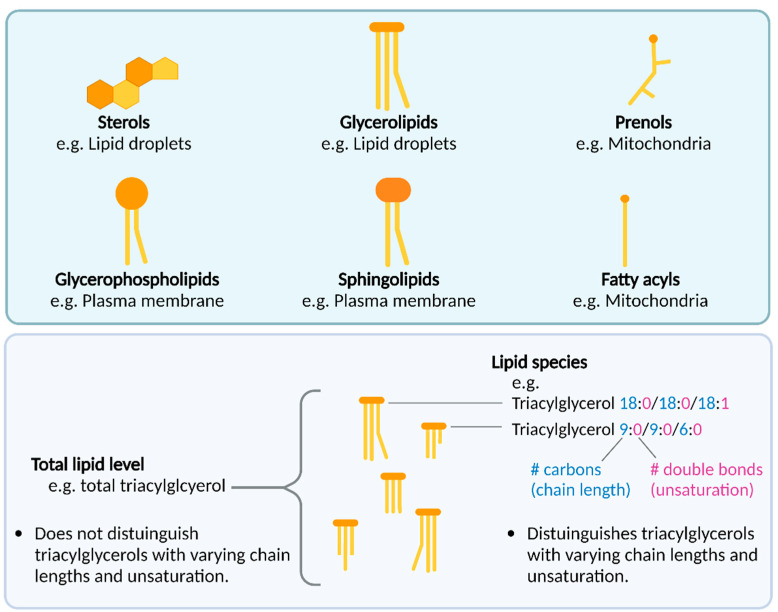
Top panel: Lipid categories and examples of their cellular location. Bottom panel: Enzymatic and chemical assays can measure total lipid levels, while mass spectrometry techniques can discern the levels of specific lipid species, which have varying degrees of unsaturation and carbon chain lengths. Created with BioRender.com (accessed on 26 October 2022).

**Figure 3 biomolecules-12-01597-f003:**
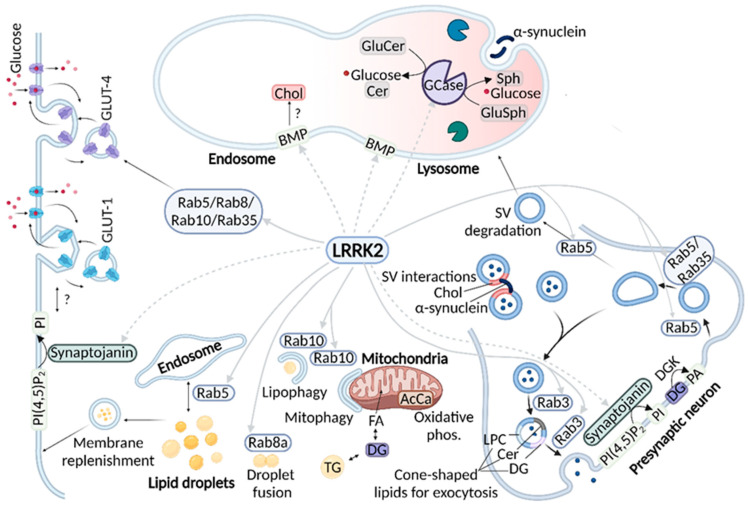
Summary of LRRK2 functions which impact lipids. Solid grey arrows represent direct phosphorylation by LRRK2 and dashed grey lines indicate a possible indirect or direct effect by LRRK2 through an unresolved mechanism of action. Synaptojanin-1 is a putative LRRK2 substrate that dephosphorylates phospholipid at the cell membrane, which is important for synaptic vesicle recycling and also insertion of GLUT-1 in the plasma membrane. Rab proteins phosphorylated by LRRK2 are also involved in recycling membrane receptors such as GLUT-4. Rabs are also involved in the trafficking of synaptic vesicles in vesicle recycling or degradation pathways. In addition, Rabs are involved in lipid droplet fusion, autophagy and transport of lipids from droplets to the cell membrane. Lipid droplets are also a reservoir for substrates important for mitochondrial oxidative phosphorylation. LRRK2 affects the endolysosomal lipid BMP, which in turn appears to affect endolysosomal cholesterol levels. LRRK2 may also have an effect on GCase activity and the levels of GCase substrates and metabolites. PI(4,5)P_2_ = phosphatidylinositol 4,5-bisphosphate, PI = phosphatidylinositol, Chol = cholesterol, BMP = bis(monoacylglycerol) phosphate, GluCer = glucosylceramide, Cer = ceramide, GalCer = galactosylceramide, Sph = sphingosine, GluSph = glucosylsphingosine, SV = synaptic vesicle, LPC = lysophosphatidylcholine, DG = diacylglycerol, PA = phosphatidic acid, FA = fatty acid, AcCa = acylcarnitine, phos = phosphorylation. Created with BioRender.com (accessed on 26 October 2022).

## Data Availability

Not applicable.

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
