# Peer review of "LRRK2 and Lipid Pathways: Implications for Parkinson’s Disease"

_biomolecules, 2022, doi:10.3390/biom12111597_

Round 1

Reviewer 1 Report

I read with interest the article by Jasmin Galper entitled " LRRK2 and lipid pathways: implications for Parkinson's disease pathology". I have some minor comments to the authors:

-        This is a well written paper and highlights an extremely relevant review on the associations between LRRK2 and lipids, and the functional consequences of LRRK2-associated lipid changes. the unique lipid signature of LRRK2 mutation carriers may be useful features to diagnostic and prognostic biomarker, and others.

-        It would be helpful to cite a reference in line 84, BMP isoforms are significantly raised in LRRK2 G2019S, and R1441G/C mutation (doi: https://doi.org/10.1101/2022.08.03.22278207).

-        Rab35 mediates LRRK2-induced different cellular functions, including endosomal trafficking, phagocytosis, cell migration and neurite outgrowth. Therefore, I suggest authors to expand their description on the associations between LRRK2 and Rab35.

-        Please add the abbreviations e.g. synaptojanin-1 in the text.

Reviewer 2 Report

The manuscript by Galper, Kim & Dzamko provides a nice overview about the link between LRRK2 and lipid physiopathology. It covers current knowledge coming from knockout as well as Parkinson's disease gain of function models, and reports data about all lipid classes. I have a few suggestions to improve the work:

- in my opinion, Figure 1 would benefit of an extra schematic summarising the major lipid classes described in paragraph 1, together with the main subcellular localisation. This could help the reader in following the flow of the next paragraphs as well, and would complement Figure 3 nicely.

- I believe the manuscript will improve if the authors were to add some extra discussion. For example: what could be the mechanism determining GCase levels in LRRK2 models? Also, it is intriguing that loss and gain of function models often show the same phenotype. What is the authors' view in this respect?

- in section 4, only 2 synaptic interactors of LRRK2 are mentioned, although many candidate partners at the pre- and post-synaptic site have been reported for LRRK2 (see for example PMID: 35011731, PMID: 32973447, PMID: 33206398). Could you please comment on that? 

Reviewer 3 Report

Galper and colleagues have written a thoughtful, comprehensive and timely review on LRRK2 and how it may regulate various aspects of lipid metabolism.  There is increasing awareness and interest in lipid biology in multiple diseases, including Parkinson’s disease so this is a topical review which I expect will be of interest to a wider audience and ultimately well cited.

The authors raise a number of key points, which perhaps could be further emphasized:

1)      Whether peripheral changes in lipids are reflected in the brain.

2)      To what extent does CSF lipid changes reflect / correlate what is happening in the brain.

3)      How different analytical methods have influenced the results / interpretations in the key publications.  Some of the technical limitations are highlighted, but more can be said, for example, on limitations of spatial and temporal resolution in measuring lipids, i.e. ideally one would measure sub-cellular compartments as the whole cell (or indeed whole tissue) analysis that are conducted in the vast majority of studies do not account for this.

There are a number of points that should be considered by the authors:

1)      Section 1.3 focuses on lipid analysis on rodent LRRK2 knockout models.  Whilst this does potentially inform on the role of this kinase in aspects of lipid metabolism in many ways the more pertinent comparators should be the pathogenic mutation knock-in models.  Figure 2 should be updated to include this.

2)      Section 1.4, lines 101-107: Can the observed changes in phosphatidylinositol and BMPs be linked mechanistically via lipid pathway analysis?

3)      Section 3.1, lines 247-249: The authors should elaborate on the magnitude / robustness of these data.  Can it be used as a biomarker, particularly as LRRK2 small molecules / ASOs are in the clinic?

4)      Section 3.2, lines 297-317: Have lipid droplets been evaluated / measured in human post-mortem brains?  If so, include the data.  If not, then highlight this gap.

5)      Section 4, lines 387-391: There is an extensive literature on PUFAs and a-synuclein, which could be addressed better here.

6)      Section 5, lines 440-443: the authors should highlight that these measurements have been made in urine and that there is not a reported difference in BMP levels in the CSF from LRRK2 mutation carriers.  This section could expand further on the utility of lipid biomarkers in current clinical trials (cf. #3, above).

Minor points:

Line 22 – change “In LRRK2 mutation carrier humans,….” to “In humans that carry LRRK2 mutations….”

Line 31 – Reference 1, West et al 2005, described the biochemical activity of LRRK2, not the genetics / frequency in disease.  Please identify a more relevant citation for this first sentence.

Line 48 – Reference 7, this is from 2007, is there a more recent review that can be cited?

Figure 3 – typo in the figure: “Mitopophagy”; line 429: replace ‘affect’ with ‘effect’.

Reviewer 4 Report

This paper examines the lipid alterations in subjects with LRRK2 mutations and LRRK2 animal models. Further, the manuscript aims to link LRRK2 and lipid-regulated systems to PD. The manuscript is well written, and the LRRK2 and lipid pathways are clearly described. Even though, the implications for Parkinson's disease (as the title states) are out of focus. The title, the abstract, and the overview section suggest that the review will focus on Parkinson's disease pathology. Still, the article's primary focus is the LRRK2 and lipid pathways. Therefore, I recommend changing the title and making it clear to the reader that the article will focus on LRRK2 and lipid pathways and not on Parkinson's disease. If the authors still want to address the role of LRRK2 and lipid pathways in Parkinson's disease, I recommend an entire section reviewing the most critical aspects of Parkinson's disease histopathological abnormalities. It would also be helpful if the authors stated the implications of LRRK2 and lipid pathways on dopaminergic cell loss in substantia nigra compacta.
